# Enhanced NO_2_ Sensing Performance of ZnO-SnO_2_ Heterojunction Derived from Metal-Organic Frameworks

**DOI:** 10.3390/nano12213726

**Published:** 2022-10-23

**Authors:** Xiaowei Ren, Ze Xu, Zhongtai Zhang, Zilong Tang

**Affiliations:** State Key Laboratory of New Ceramics and Fine Processing, School of Materials Science and Engineering, Tsinghua University, Beijing 100084, China

**Keywords:** NO_2_ gas sensor, ZnO, SnO_2_, heterojunction, MOFs derivatives

## Abstract

Nitrogen dioxide (NO_2_) is the major reason for acid rain and respiratory illness in humans. Therefore, rapid, portable, and effective detection of NO_2_ is essential. Herein, a novel and simple method to construct a ZnO-SnO_2_ heterojunction is fabricated by pyrolysis of bimetallic metal organic frameworks. The sensitivity of ZnO-SnO_2_ heterojunction towards 0.2 ppm NO_2_ under 180 °C is 37, which is 3 times that of pure ZnO and SnO_2_. The construction of heterojunction speeds up the response-recovery process, and this kind of material exhibits lower detection limit. The construction of heterojunction can significantly improve the NO_2_ sensitivity. The selectivity, stability, and moisture resistance of ZnO-SnO_2_ heterojunction are carried out. This could enable the realization of highly selective and sensitive portable detection of NO_2_.

## 1. Introduction

Air pollution is one of the greatest threaten to environment and human health. Outdoor air pollution produced by human activities, including road traffic, industrial emission and residential heating [1]. Exposure to NO_2_ increased the risk of laryngo-tracheo-bronchitis related disease, mostly due to influenza viral infections [2]. According to WHO Global air quality guidelines [3], the thresholds of several major air pollutants are defined, among which, the guideline value of NO_2_ is 25 μg/m^3^ (at standard atmospheric pressure, 20 °C, 25 μg/m^3^ ≈ 13 ppb). Up to now, it is still a major challenge to achieve rapid detection to ppb level NO_2_. Therefore, the development of NO_2_ gas sensors that can respond quickly to low concentrations is of great scientific significance and application value.

At present, much research is focused on enhancing the sensitivity and reducing the detection limit, including microstructure design to improve specific surface area, defect regulation to enhance adsorption performance, facet control to increase effective adsorption sites, loading precious metals and constructing heterostructure to promote sensitivity [4,5,6,7,8,9]. Among these modification methods, combining two sensitive metal oxides to construct n-n or n-p heterojunction has been proved to be an effective way to enhance the sensitivity, due to the abundant sensitive candidate components to choose from [10]. Many methods are reported to synthesize metal oxides heterojunction [6,11], the pyrolysis of bimetallic metal organic frameworks (MOFs) precursors to obtain metal oxides composite structure has recently been regarded as an effective method [12]. MOFs are known as porous coordination polymers, constructed by metal-based nodes bridged with organic ligand [13]. Recently, much research has focused on the preparation of porous nanomaterials by using MOF as self-sacrificing template [14,15,16,17]. Woo et al. [18] developed ZnO/CNTF composites by calcination of ZIF-8/CNTF; the fabricated gas sensor exhibited improved NO_2_ sensitivity (1.086 towards 20 ppm NO_2_ at 100 °C). Deng et al. [19] obtained laminar SnO_2_ by pyrolysis of Sn-MOF@SnO_2_, as synthesized sensors exhibited excellent formaldehyde sensing properties (10,000 to 10 ppm formaldehyde at 120 °C). Existing research results have indicated that MOF-derivatives exhibits high specific surface area and connected channel, which can improve gas accessibility and increase active sites. In the field of sensing, extensive work has proved that mixed-MOF exhibit better performance than single metal node MOFs due to the synergetic effect of diverse valence metals [20,21]. It is well known that ZnO and SnO_2_ are considered as two typical gas sensitive materials with excellent properties [22,23]. Although plenty of research works have considered bimetallic MOFs derivations, there has been research on those containing tin oxides. Therefore, the construction of ZnO-SnO_2_ heterojunction derived from bimetallic MOFs is expected to improve the gas sensing performance.

Herein, we report a simple strategy to obtain metal oxides heterojunction. At first, the Zn/Sn-bimetallic MOFs are directly synthesized by one-pot methods, and the ZnO-SnO_2_ heterojunction is obtained by thermal decomposition subsequently. The as-prepared ZnO-SnO_2_ composites derived from bimetallic MOFs exhibits excellent sensitivity to ppb-level NO_2_. This study provides a new perspective for the design of metal oxides heterojunction.

## 2. Experimental

### 2.1. Synthesis of Zn/Sn Bimetallic MOF

Two Zn/Sn bimetallic MOFs with different Zn/Sn ratio were synthesized. At first, 1.2 g Zinc nitrate (≥99%, Zn(NO_3_)_2_•6H_2_O, Beijing Modern Oriental Fine Chemistry Co., Ltd., Beijing, China) was dissolved in 40 mL N, N-Dimethylformamide (≥99.5%, DMF, HCON(CH_3_)_2_, Shanghai Titan Scientific Co., Ltd., Shanghai, China). Then Terephthalic Acid (99%, C_8_H_6_O_4_, Shanghai Titan Scientific Co., Ltd., Shanghai, China) with different weight (1 g, 1.334 g) was added to above solution. Tin (II) chloride (≥99%, SnCl_2_, Shanghai Macklin Biochemical Technology Co., Ltd., Shanghai, China) with different molar ratio to Zinc was added to above solution (2 mM, 4 mM). 1.2 mL Triethylamine (99%, C_6_H_15_N, Shanghai Titan Scientific Co., Ltd., Shanghai, China) was dropped into the solution after 20 min stirring. Precipitate immediately began to appear in the solution. After 40 min stirring, the products were separated by centrifuged with DMF. Finally, the Zn/Sn(II)-MOF was dried in vacuum oven at 80 °C for 16 h.

### 2.2. Synthesis of Zn-BDC MOF and Sn-BDC MOF

For comparison, the pure Zn-MOF and Sn-MOF were also synthesized by above steps. For Zn-MOF, 1.2 g Zinc nitrate and 0.667 g Terephthalic Acid were added. For Sn-MOF, 0.76 g SnCl_2_ and 0.667 g Terephthalic Acid were added. The other steps were the same as above.

### 2.3. Synthesis of ZnO-SnO_2_ Metal Oxides

The two metal oxides were obtained by thermal decomposition of Zn/Sn(II)-MOF. A suitable amount of the precursor was pyrolysis in air at 800 °C for 2 h at a heating rate of 5 °C/min. The final products with good crystallinity were obtained.

### 2.4. Preparation of Gas Sensor and Sensing Measurements

A volume of 100 μL methanol was added to the 25 mg sample and the suspension was mixed evenly by ultrasonic for 1 h. Then, the sensing slurry was drop-coated on the Au interdigital electrodes with a spacing of 0.2 mm. After that, the interdigital electrodes were dried at 60 °C overnight.

Gas sensing measurement was carried out in a computer-controlled evaluation system. The interdigital electrode is placed on the heating table in the test chamber (~0.02 L), and the heating table has heating and temperature detection device. The real time dynamic resistance of the fabricated sensors was recorded by a digital multimeter (Agilent 34465A). The dry testing gases involved in this research (10 ppm NO_2_, 1000 ppm H_2_S, 100 ppm NH_3_, 500 ppm CO, 200 ppm C_2_H_5_OH, all balanced by N_2_) were purchased from Beijing AP BAIF Gases Industry Co., Ltd., Beijing, China. The concentration of target gas was controlled by regulating the flow rates of each gas by mass flow controllers (MFC, Beijing Qixing Co., Ltd., Beijing, China), and total flow rate was settled at 200 mL/min. Specifically, the 10 ppm NO_2_ with flow rate of 2 mL/min diluted by 156 mL/min dry N_2_ and 42 mL/min dry O_2_ was injected to the chamber to obtain the desired concentration of 0.1 ppm NO_2_. The gas response was defined as *R*_g_/*R*_a_ for oxidizing gas and *R*_a_/*R*_g_ for reducing gas, where *R*_g_ is the resistance in target gas and *R*_a_ is the resistance in air. Humidity tests were measured by passing the flow gas through a quartz flask filled with saturated solutions of K_2_CO_3_ and pure deionized water to attain the relative humidity (RH) levels of 43% ± 2% and 90% ± 2%, a humidity sensor probe was installed at the outlet of the quartz flask, more testing details described in the Appendix A. Except for the humidity test, other gas performance tests were carried out under dry conditions.

### 2.5. Characterization

Crystal structure of the samples are recorded by powder X-ray diffraction (XRD, Cu Kα, Rigaku D/max-V2500, Rigaku Co., Ltd., Tokyo, Japan). The microstructure and morphology were obtained by using a scanning electron microscope (SEM, Carl Zeiss, Oberkochen, Germany) and transmission electron microscope (TEM, JEOL JEM-2100F, Tokyo, Japan). The elemental valence and surface electronic structure of the samples were analyzed by X-ray photoelectron spectroscopy (XPS, ESCALAB 250Xi, Thermo Fisher Scientific, Waltham, MA, USA). The adsorption characteristic was obtained by UV-vis-NIR (Lambda 950, Perkin-Elmer, Waltham, MA, USA) spectrophotometer. Raman spectra were measured by LabRAM HR800 Raman spectrometer under the excitation of 633 nm laser. N_2_ adsorption/desorption isotherms at 77 K were carried out by using an American Contador QuadraSorb SI-MP instrument and the specific surface area was calculated from N_2_ desorption isotherms by using the Brunauer-Emmett-Teller (BET) method.

## 3. Results and Discussion

### 3.1. Material Characterization

The formation of metal oxide heterojunction is illustrated in Figure 1. For simplicity, the samples were abbreviated to S1, S2, S3, and S4, where S1 stands for the metal oxides derived from Zn-MOF, S4 is the derivation of Sn-MOF, S2 and S3 are derivations of Zn/Sn bimetallic MOF with different ratio of Zn to Sn. The structure of the precursors is characterized by XRD, as shown in Appendix A. The diffraction peaks of pure Zn-MOF and Sn-MOF are similar with previous literature [24,25]. The Zn/Sn bimetallic MOF exhibits unique diffraction pattern from Zn-MOF and Sn-MOF, which demonstrated that the bimetallic MOF are not simply mechanically mixed in the material, but have some different arrangements [26]. The contents of zinc element and tin element of sample S2 and S3 are also determined by EDS, as shown in Appendix A, indicating that sample S2 and S3 are composites of ZnO and SnO_2_, and the ratio of Zn to Sn is approximately 1:0.3 and 1:2, respectively.

The X-ray diffraction results of the samples are shown in Figure 2a. The main diffraction peaks of the sample without tin element are well indexed with ZnO (PDF# 99-0111) of wurtzite structure. With the addition of tin element, the diffraction of SnO_2_ (PDF# 77-0452) appears in sample S2 and S3. The diffraction peak of sample S4 matches well with the characteristic of SnO_2_. The diffraction peaks of all samples match the characteristic peaks contained in either or both ZnO and SnO_2_, and no other peaks were observed. Figure 2b exhibits the heterojunction of ZnO-SnO_2_ composites. The fringe spacing of grains on both sides of the interface is obviously different. The fringe spacing of the left grain is 2.8 Å, which can only correspond to the (100) plane of ZnO. The fringe spacing of the right grain is 3.4 Å, which can only match with the (110) plane of SnO_2_. The well contact of two grains with different composition confirms the existence of heterogeneous structure. Figure 2d is an enlarged image of the red selection in Figure 2c. Nanoparticles grow on the surface of hexagonal nanosheets, the clear lattice interplanar distance of nanoparticles is 3.4 Å, assigned to the (110) planed of tetragonal SnO_2_. The hexagonal nanosheets with the lattice fringe of 2.8 Å are consistent with (100) planes of wurtzite ZnO. From the selected area electron diffraction (SAED) pattern in Figure 2e, diffraction rings with (101) and (100) planes of ZnO and (110) and (101) planes of SnO_2_ can be observed.

The element distribution of sample S2 can be concluded by EDS map in Figure 2f–i. In agreement with the previous results of XRD and HRTEM, the region with hexagonal sheet structure is dominated by ZnO, small particles attached to the hexagon are ZnO and SnO_2_. The results of EDS are in good agreement with XRD results. During the heat treatment, bimetallic MOFs begin to fall apart, and the metal ions are reoriented and oxidized to two kinds of metal oxides, and the n-n heterojunction structure is eventually formed. TEM and SEM images and corresponding grain size analysis of sample S1–S4 are shown in Appendix A. The mean grain size of S1 is 107 nm. The grain size of sample S2 is heterogeneous, consisting of two grains with different sizes; therefore, the average grain size is calculated according to two different intervals. The mean size of small grain and large grain is 20 nm and 193 nm, respectively. As described above, the large particles in the ZnO-SnO_2_ composites are mainly ZnO, and the small particles are mainly SnO_2_. The grain size distribution of S3 is similar with S2. In sample S4 (SnO_2_), part of large particles condensed into thick sheets after pyrolysis at 800 °C. No clumps of condensation were found in the S2 and S3, indicating that the presence of ZnO inhibits the agglomeration and growth of SnO_2_.

X-ray photoelectron spectroscopy is carried out on the sample S1, S2 and S4 to determine the effect of heterojunction on the surface chemical state of Zn, Sn and O. All the peaks are calibrated by C 1*s* (284.8 eV). The high-resolution spectrum of Zn 2*p* in sample S1 and S2 exhibited in Figure 3a can be deconvoluted to two characteristic peaks correspond to Zn 2*p*_3/2_ and Zn 2*p*_1/2_, indicating +2 valance of Zn in the samples. Comparing the Zn 2*p* peaks in S1, the Zn 2*p* peaks of S2 shift 0.5 eV to low binding energy. The shifts of binding energy could be attributed to the different electronegativity of Zn ions [27], which could be ascribed to the electron transfer between ZnO and SnO_2_ in S2. Figure 3b presents the characteristic peaks of Sn 3*d*_3/2_ and Sn 3*d*_5/3_ located at 494.8 eV and 486.4 eV, respectively, indicating the +4 valance of Sn. The Sn 3*d* peaks of S2 shift 0.2 eV to high binding energy compared with S4, which again confirms the electron transfer between ZnO and SnO_2_ in S2. As shown in Appendix A–d, the asymmetric O 1*s* peak of S1, S2 and S4 can be separated into two peaks, the peak located at 530.2 eV is related to lattice oxygen (O_L_), and another located at 531.4~531.6 eV is associated with surface adsorbed oxygen (O_C_), including physically-adsorbed, chemisorbed, dissociated, and hydroxy-related oxygen). The latter is usually closely related to the properties of gas-sensitive materials, and the sample S2 has more surface adsorbed oxygen.

The specific surface area and pore size distribution are critical to gas sensitive materials. The N_2_ adsorption/desorption isotherms of sample S1–S4 exhibit in Figure 4a. S2 and S3 exhibit IV type characteristics, demonstrating the mesoporous structure of the material [15]. The BET specific surface area of S1–S4 are 27.03, 29.73, 35.70, 21.92 m^2^/g, respectively. High specific surface area can expose more active sites, which is conducive to the adsorption of gas molecules.

In order to investigate more detailed structure information, Raman spectroscopy was carried out to characterize the structure information and molecular vibrations of the materials. As shown in Figure 4b, the characteristic peaks of ZnO centered at around 334, 381 and 439 cm^−1^ [28], corresponding to the vibration modes of *E*_2_ (high) − *E*_2_ (low), *A*_1T_, and *E*_2_ (high), respectively. The peaks at 635 and 778 cm^−1^ corresponds to *A*_1g_ and *B*_2g_ modes, respectively, which are attributed to the stretching of Sn–O bonds [29]. The main peaks of all samples can correspond to the characteristic peaks of ZnO or SnO_2_, which is consistent with previous analysis. Furthermore, the spectrum of Raman mapping was obtained by a 50 × 50 μm area as shown in Appendix A, providing the spatial information of ZnO and SnO_2_ phases in micro level.

The variation of optical absorption characteristic and band gap energies of S1, S2, and S4 samples were analyzed by UV–vis diffuse reflectance spectroscopy in the wavelength range 250–800 nm. As shown in Figure 5a, the sample ZnO exhibits an intense adsorption edge around λ < 380 nm assigned by the intrinsic wide band gap. The absorption edge of SnO_2_ is approximately located at 300 nm. Two absorption bands can be observed in S2. The band gap energies of S1, S2 and S4 were determined by Tauc-plot methods [30]. According to previous study, materials involved in this paper are direct band gap [31,32]. The estimated band gaps were 3.19, 3.22, and 3.63 eV for S1, S2 and S4, respectively. The appearance of two absorption bands and the calculated band gap energy of S2 confirmed that the combination of ZnO and SnO_2_. Based on the analysis result of TEM, XRD, Raman, XPS and UV–vis, the formation of ZnO-SnO_2_ heterojunction was confirmed, which could have a particular influence on the physical and chemical properties of the materials.

### 3.2. Gas Sensing Properties

For metal oxides semiconductors, the adsorption characteristic of gas on the surface of materials are strongly affected by temperature. Firstly, the sensing performance of all samples was carried out to 0.2 ppm NO_2_ at different working temperature, as shown in Figure 6a. The sensitivity firstly increases to optimal value and then decreases with the increase of temperature. NO_2_ molecules do not have enough activation energy to react with the adsorbed oxygen species at low temperature. Rising temperature can promote the physical and chemical adsorption of NO_2_. The desorption rate accelerates when the temperature rises to a certain extent, leading to the decrease of response value. The resistance was too large to measure for sample S3 contacted with NO_2_ at 160 °C. Although S3 sample has the highest response value at 180 °C, the excessive resistance is not conducive to practical application, hence, sample S2 is selected as the research object. Compared with pure ZnO (S1) or SnO_2_ (S4), the samples combined with ZnO and SnO_2_ exhibit better NO_2_ sensing performance. For sample S2, the NO_2_ sensing sensitivity is about 37 at 180 °C, which is 3 times that of ZnO (S1) and SnO_2_ (S4). This demonstrating the proper combination of ZnO and SnO_2_ can improve the NO_2_ sensing performance. Figure 6b illustrates the reproducibility of S2 to 0.2 ppm NO_2_ at 180 °C under dry condition, demonstrating the stability and the n-type semiconducting conduction characteristic of the as fabricated gas sensor. Figure 6c exhibits the dynamic resistance curve towards 0.02–5 ppm NO_2_ under 180 °C, resistance increases with the increase of NO_2_ concentration. The actual response value is 1.21 towards 20 ppb NO_2_, which means the gas sensors fabricated in this paper has a very low detection limit, the dynamic resistance towards 20–50 ppb NO_2_ was shown in Appendix A. The fabricated gas sensor exhibits a significant response to ppb-level NO_2_. The relationship of response value and NO_2_ concentration are exhibited in Appendix A, blue line is the fitting curve, and there are different fitting curves at low concentration and high concentration. Figure 6d presents the baseline resistance as a function of temperature for all samples. With the increase of temperature, the baseline resistance decreases. At 180 °C, baseline resistance (*R*_a_) is ranked from low to high: S1 < S2 < S3 < S4. S4 exhibits the highest resistance, indicating the greatest activation energy is required, and S3 is next. With the increase of Sn content, the activation energy increases. In addition, the baseline resistance also influenced by the grain size and contact interface. The resistance increase in S2 and S3 can be ascribed to the increase of heterogeneous interface. From Figure 6e it can be concluded that the combination of ZnO and SnO_2_ greatly improve the NO_2_ sensitivity and accelerate the response-recovery time. The response values of S1, S2, S3, and S4 towards 0.1–5 ppm NO_2_ at 180 °C are exhibited in Figure 6f. The NO_2_ sensitivity of S2 and S3 is much higher than the single component either ZnO or SnO_2_.

Stability is another important index for evaluating gas sensors. Figure 7a illustrates that the fabricated gas sensor exhibits excellent repeatability after several days, even if no degradation can be observed, which may be due to the stability of the microstructure of the materials after high temperature heat treatment. The effect of humidity on the NO_2_ sensing response was investigated under 43% RH and 90% RH condition, as shown in Figure 7b. In the presence of water molecules, the sensor presented better response-recovery characteristic. The response value to NO_2_ decreased, indicating that water molecules will hinder the adsorption of NO_2_ gas molecules.

Figure 7c illustrates the sensitivity of fabricated gas sensor towards 0.2 ppm NO_2_ and other higher concentration gases (50 ppm H_2_S, 50 ppm CO, 40 ppm C_2_H_5_OH, 50 ppm NH_3_) at 180 °C. The selectivity of NO_2_ on sensing materials is affected by several reasons, including electron affinity of target gas and the amount of adsorbed gas molecules on the surface of sensing materials under optimal working temperature, as well as surface defect species [33,34]. The mechanism of the selective detection of ZnO-SnO_2_ composition to NO_2_ over other gases can be summed up to three points. Firstly, the electron affinity of NO_2_ molecules is higher than other interfering gases, which means NO_2_ is more inclined to grab electrons from the conduction band of material. Secondly, operating temperature also has a significant influence on the selective adsorption of gas. The work temperature in this paper is around 180 °C, which may be suitable for NO_2_ adsorption. Thirdly, it has been proved that ZnO and SnO_2_ are two typical NO_2_ adsorption material, the combination of the two sensitive materials will produce a synergistic effect and enhance the NO_2_ sensitive characteristics [35].

To further understand the practicality of the gas sensor fabricated by ZnO-SnO_2_ in this study, the NO_2_ sensing performance were compared with ZnO-based, SnO_2_-based and ZnO-SnO_2_ sensors reported recently. As shown in Table 1, sample S2 exhibits relatively sensitive performance towards NO_2_.

### 3.3. NO_2_ Sensing Mechanism

After the above analysis of the structure and gas sensitive performance of ZnO, SnO_2_ and ZnO-SnO_2_ samples, it can be concluded that the construction of ZnO-SnO_2_ heterojunction can significantly improve the gas sensitivity of NO_2_, the n-n heterojunction plays a vital role in the sensing properties. Figure 8 illustrates the band structure of ZnO-SnO_2_ heterojunction. The band structures of ZnO and SnO_2_ that were previously in contact with each other are exhibited in Figure 8a. The *qφ* (work function) and *E*_g_ (band gap) of ZnO are 5.2 eV, 3.19 eV, respectively, and those value of SnO_2_ are 4.55 eV, 3.63 eV, respectively. The heterojunction formed at the interface between ZnO and SnO_2_. The electrons will flow from SnO_2_ to ZnO owing to the lower work function of SnO_2_ compared with ZnO [45], leading to the increasing number of free electrons on the surface of ZnO. The Femi levels will become equal when a dynamic equilibrium is established between ZnO and SnO_2_. Therefore, the electron depletion layer formed at the SnO_2_ side, and electron accumulation layer formed at the ZnO side. The shift of binding energy in XPS analysis confirmed the different electronegativity of the elements. Besides, NO_2_ is an oxidizing gas. The formation of electron accumulation layer will further promote NO_2_ to grab electrons from the conduction band of ZnO. In addition, increasing number of electrons will facilitate more oxygen species adsorbed on the surface of ZnO and further reacts with NO_2_ simultaneously. The corresponding equation is shown below [46]:NO_2_ + e^−^ → NO_2_^−^(1)
NO_2_^−^ + O^−^ + 2e^−^ → NO + 2O^2−^(2)

Generally, the sensing performance of metal oxides semiconductors can be affected by potential barriers. The interfaces between ZnO-ZnO, ZnO-SnO_2_, and SnO_2_-SnO_2_ may have an important influence on the sensing performance [41]. The smaller particles of SnO_2_ in ZnO-SnO_2_ composition are facilitates NO_2_ adsorption. Therefore, the excellent NO_2_ sensing performance can contribute to the increased free electrons, larger number of adsorption sites, and the interfaces effect of ZnO-SnO_2_ heterojunction.

A high performance gas sensor fabricated by sensitive materials will be widely used in environmental monitoring, artificial intelligence, the Internet of Things, and wearable devices in the future [47,48,49].

## 4. Conclusions

In summary, bimetallic MOFs were synthesized by direct solution methods, and the ZnO-SnO_2_ metal oxides heterojunction structure was obtained after subsequent heat treatment. The size of SnO_2_ nanoparticles is 5–30 nm, and the diameter of ZnO hexagonal nanoparticles are 0.1–2 μm. The construction of ZnO-SnO_2_ heterojunction greatly enhanced the NO_2_ sensing performance. When the molar ratio of ZnO to SnO_2_ is 1:0.3, the response value of ZnO-SnO_2_ heterostructures is 37 towards 0.2 ppm NO_2_ at 180 °C, which is 3 times higher than pure ZnO and SnO_2_ made by the same methods. Furthermore, ZnO-SnO_2_ heterostructures could detect NO_2_ as low as 20 ppb. Moreover, the ZnO-SnO_2_ heterostructure exhibits excellent selectivity to NO_2_. The outstanding NO_2_ sensing performance of ZnO-SnO_2_ heterostructure can be attributed to the synergistic effect of n-n heterojunction between ZnO-SnO_2_.

## Figures and Tables

**Figure 1 nanomaterials-12-03726-f001:**
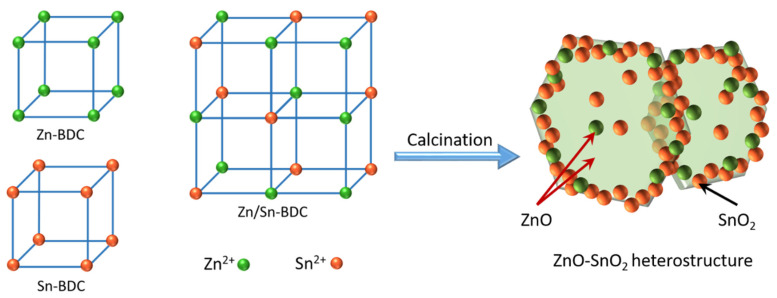
Schematic diagram of heterogeneous bonding of bimetallic oxides.

**Figure 2 nanomaterials-12-03726-f002:**
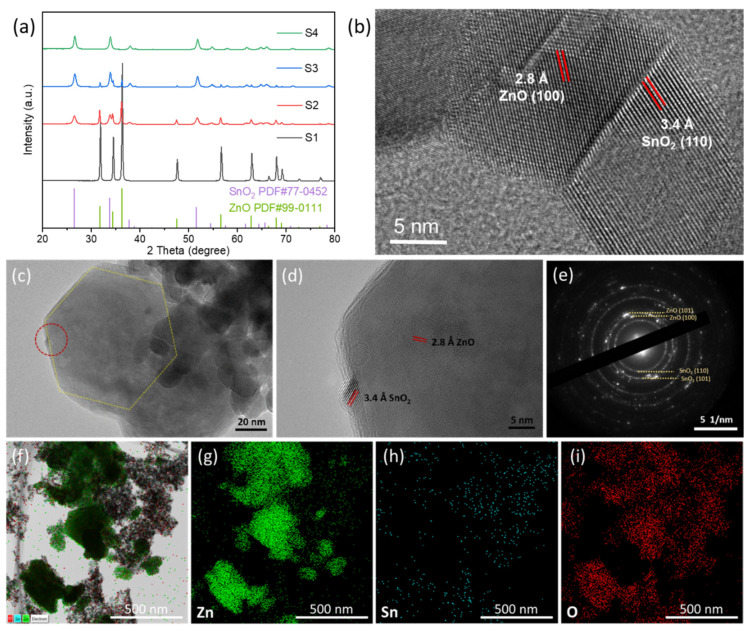
(**a**) XRD patterns of Zn-Sn bimetallic derivatives. (**b**–**d**) HRTEM, (**e**) SAED images of S2. (**f–i**) TEM image of S2 and corresponding elemental mapping for Zn, Sn and O elements.

**Figure 3 nanomaterials-12-03726-f003:**
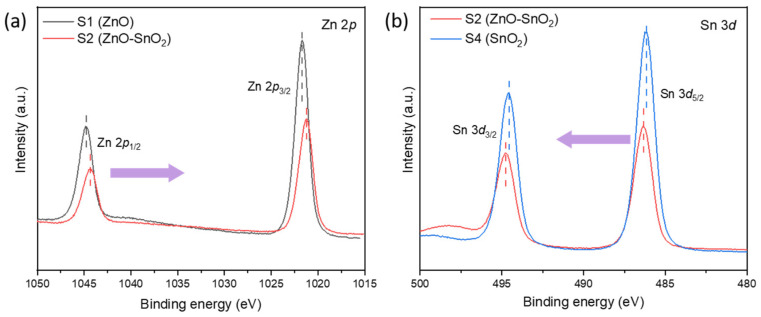
XPS curve of (**a**) Zn 2*p* for sample S1 and S2, (**b**) Sn 3*d* for sample S2 and S4.

**Figure 4 nanomaterials-12-03726-f004:**
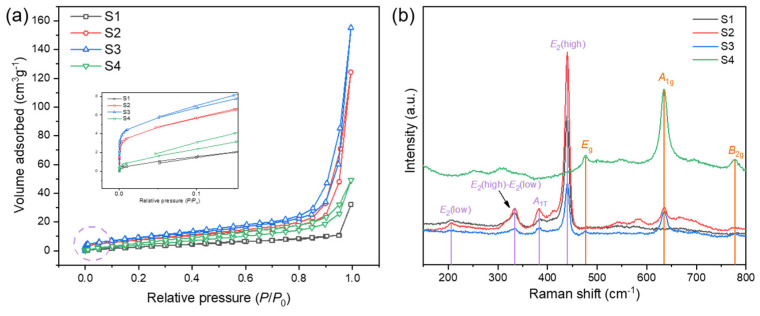
(**a**) The nitrogen adsorption/desorption isotherm of S1–S4 and the curve at low pressure zone (inset) (**b**) Raman spectrum of all samples (purple letters represent characteristic peaks of ZnO and orange represents SnO_2_).

**Figure 5 nanomaterials-12-03726-f005:**
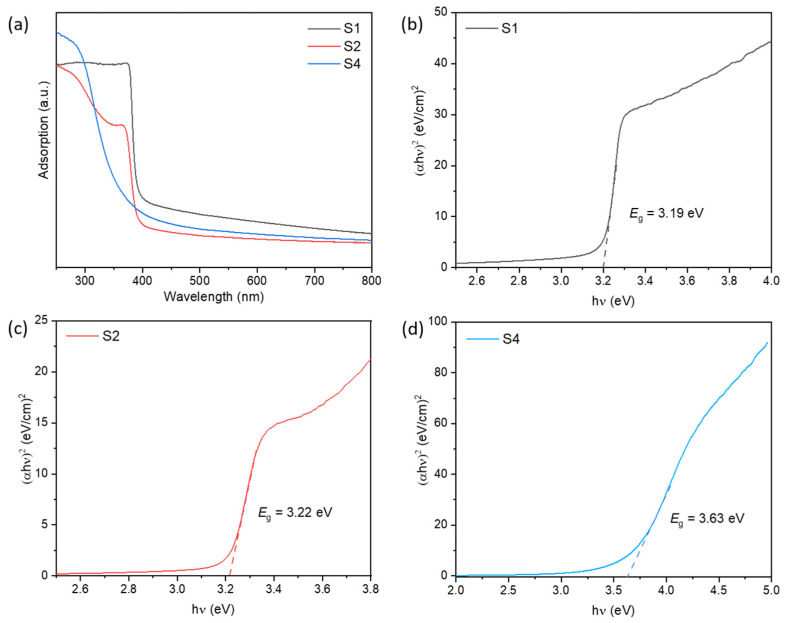
(**a**) UV–vis diffuse reflectance spectra of S1, S2 and S4. The energy band gap of (**b**) S1, (**c**) S2 and (**d**) S4 calculated by Tauc-plot methods.

**Figure 6 nanomaterials-12-03726-f006:**
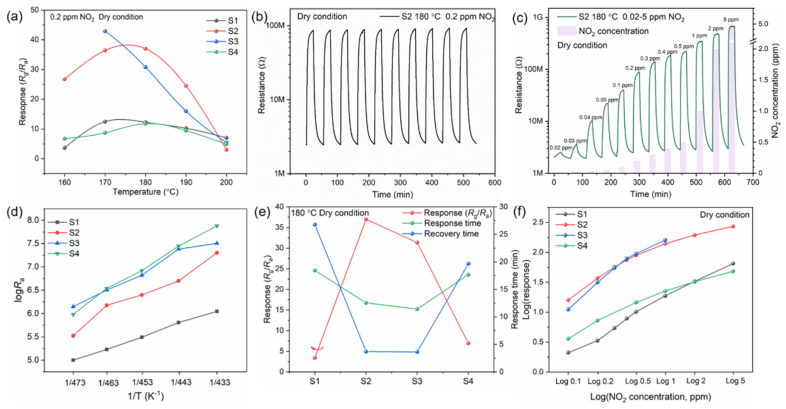
(**a**) Response value of the sensors towards 0.2 ppm NO_2_ at different working temperature. (**b**) Reproducibility of S2 to 0.2 ppm NO_2_ at 180 °C. (**c**) Dynamic response of S2 to 0.02–5 ppm NO_2_ at 180 °C. (**d**) *R*_a_ of the sensors at different operating temperature. (**e**) Response value and response and recovery time of sample S1, S2, S3 and S4 towards 0.2 ppm NO_2_ at 180 °C. (**f**) Response value of all samples to 0.1–5 ppmNO_2_ at 180 °C. (Double logarithmic coordinate axis was used).

**Figure 7 nanomaterials-12-03726-f007:**
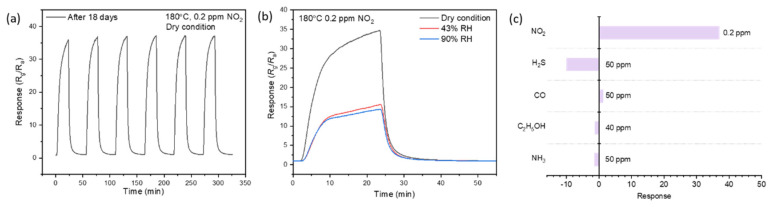
(**a**) Cyclic response curve of S2 after 18 days. (**b**) Response curve of S2 to 0.2 ppm NO_2_ under dry and humidity condition. (**c**) The response value of S2 to 0.2 ppm NO_2_ and 40–50 ppm other gases at 180 °C under dry condition (Negative values represent p–type responses).

**Figure 8 nanomaterials-12-03726-f008:**
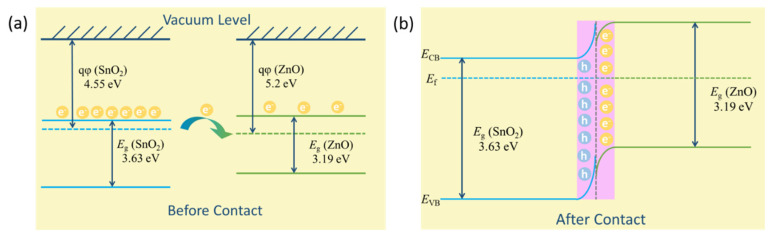
Schematic diagram for the band structure of ZnO-SnO_2_ (**a**) before contact, (**b**) after contact.

**Table 1 nanomaterials-12-03726-t001:** NO_2_ sensing performance of the materials synthesized in this study and other materials in the reported literature.

Materials	NO_2_ (ppm)	Operating Temperature (°C)	Response (*R*_g_/*R*_a_ or *R*_a_/*R*_g_)	Ref.
ZnO nanoneedles	5	195	6	[36]
SnO_2_-Sn_3_O_4_ heterostructure	0.2	150	11	[37]
ZnO@rGO	10	RT	6.77	[38]
MXene sphere/ZnO	100	RT	1.72	[39]
Sputtered SnO_2_/ZnO	5	100	26.4	[40]
ZnO-SnO_2_ hetero-nanowires	0.25	150	17	[41]
ZnO–SnO_2_ composite	1	350	21.5	[42]
ZnO nanorod/SnO_2_ film	50	300	1.37	[43]
SnO_2_@ZnO	1	150	25	[44]
ZnO-SnO_2_ heterojunction	0.2	180	37	this work

The definitions are the same in this article, and some values were read from the figures in references, which may not be very accurate.

## Data Availability

The data that support the findings of this study are available from the corresponding author upon reasonable request.

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
