# Peer review of "Enhanced NO2 Sensing Performance of ZnO-SnO2 Heterojunction Derived from Metal-Organic Frameworks"

_nanomaterials, 2022, doi:10.3390/nano12213726_

Round 1
Reviewer 1 Report
Despite the fact that the experiment was done at a good level, the results are a priori expected, and the main disadvantage of the article is the weak novelty. However, I believe that the article can be accepted for publication after revision, since the proposed synthesis method made it possible to obtain a material with a large specific surface area after high-temperature annealing at 800 C.
The authors should correct the following:
1. From diffractograms, it is necessary to estimate the size of crystallites for all samples (oxides) and discuss how the change in composition affects this parameter.
2. Similarly, it is necessary to provide information about the specific surface area for all samples (not only for S2). This will allow a more motivated discussion of what is responsible for increasing the sensitivity of composites compared to pure oxides - namely electron transfer or a simple increase in surface area. It will also be useful to normalize the sensor signal by the specific surface area, this will reveal the influence of the composition on the sensor characteristics.
3. Fig. 6d needs to be redrown in coordinates lgR vs 1/T (K-1). This will allow discussing the influence of microstructure and composition parameters on the conductivity of samples, in particular, the activation/non-activation conductivity mechanism, the values of activation energy.
4. Fig. 6f needs to be redrown in double logarithmic coordinates. It is necessary to discuss the effect of the sample's composition on their sensitivity (the slope of the obtained dependencies).
All this will allow the authors to discuss more correctly the change in the sensor properties of synthesized nanocomposites in terms of the n-n heterojunction formation.
Reviewer 2 Report
The authors reported the preparation of ZnO-SnO2 heterojunction derived from bimetallic organic frameworks for high-performance NO2 sensing. The designed heterojunction structures demonstrate effectiveness for improving the sensing property, which has reference significance for future study of this area. So I recommend its publication in Nanomaterials after clarifying the following problems:
1. The heterojunction structure is the innovation and key point to fulfill the outstanding sensing property. But the formation of the heterojunction structures lacks key evidence.
2. For the experiment design, only two groups of mixed metallic oxides are designed and obtained, then how to achieve an optimized property with only two groups’ results. At least three groups are needed to judge the change tendency of properties.
3. In the introduction part, I recommend the authors add some description of the advances of ZnO and/or SnO2-based nanomaterials for NO2 sensing, especially those obtained derived from MOF
4. According to the authors’ design as shown in Fig 1, the bimetallic organic frameworks should be a one-phase material. Actually, the obtained products as shown in SEM are mixed structures with two morphologies. What is the reason for that? In that case, the sensing property may be greatly decreased due to reduced heterojunction structures.
5. For the XPS results in Fig3, the authors should check the simulation results of Fig 3e. According to their design, the formation of heterojunctions may change the binding energy of Zn-O and Sn-O bonds in the mixed metal oxides. However, in Fig 3e, the binding energies of lattice oxygen are very similar. Does that mean the amount of heterojunctions is very few?
6. The portable is one development trend of sensing devices. Is it possible for the materials in this work to combine with super-flexible even foldable substrates to form super-flexible and wearable sensing devices, such as DOI: 10.1016/j.matt.2021.07.021; 10.1007/s42765-022-00162-7? The authors can analyze that in the conclusion part to promote its future development.
7. There are many grammar errors including the misuse of singular and plural, and lack of conjunctions between two short sentences. For example, line 33, “load… construct...” should be loading and constructing; line 37, synthesis should be synthesize; line 42, “, existing” should be “, and existing”. Please check throughout the manuscript and correct them.
Round 2
Reviewer 1 Report
The article can be accepted in present form and published after careful correction of English
Reviewer 2 Report
After revision, the questions have been well solved and the quality of the manuscript has been greatly improved. So I recommend its publication in nanomaterials.